# Sol-Gel Coatings for Subaquatic Self-Cleaning Windows

**Andrew I. M. Greer [1],\*** , **David Moodie [2]**, **Graham Kerr [1]** and **Nikolaj Gadegaard [1]**

[1]   Bio-Interface Group, School of Engineering, University of Glasgow, Glasgow G12 8LT, UK; graham.kerr@fmcti.com (G.K.); Nikolaj.Gadegaard@Glasgow.ac.uk (N.G.)

[2]   Optoelectronics Group, FMC Technologies Inc, Strathclyde Business Park, Lanarkshire ML4 3PE, UK; David.Moodie@fmcti.com

\*   Correspondence: Andrew.Greer@Glasgow.ac.uk

**Abstract:** Self-cleaning windows are well known for their ability to function with airborne pollutants, but there is a growing industry for semi-permanent subaquatic optical devices, where the performance of such windows should be considered. Here sol-gel technology is explored as a means of producing self-cleaning, subaquatic, sapphire windows. We demonstrate removal of marine bacteria and, in the worst-case contamination scenario, dead North Sea crude oil (API 35). This greasy contaminant was smeared across the windows to effectively reduce optical transmission strength to just 54%. The titania-based sol-gel-coated windows can restore transmission to within 10% of the clean value in less than one day, unlike standard sapphire windows, which lose 68% transmission following contamination and aquatic submergence over the same duration. A range of theories to enhance the self-cleaning performance of the sol-gel coating were explored, but none of the tested variables were able to provide any enhancement for subaquatic performance.

**Keywords:** sol-gel; oleophobic; submarine; self-cleaning; photocatalytic

---

## 1. Introduction

$TiO_2$ in the anatase polymorph is a semi-conductor with a band gap energy of 3.2 eV, corresponding to a wavelength of 390 nm [1]. Hence, UV light (365–390 nm) has more than the required energy to excite charge transport in anatase. Positive charges will oxidize (remove) electrons from any organic matter (dirt) on the surface, and negative charges will combine with atmospheric oxygen to create radicals. The oxygen radicals and oxidized dirt react to release $CO_2$ and $H_2O$, ultimately cleaning the $TiO_2$ surface [2,3]. This self-cleaning process is well established, with window manufacturers presently implementing the technology on domestic products [4], but this is the first example of the technology being applied to sapphire windows for subaquatic applications.

The motivation for testing the windows under water is due to the increasing range of subaquatic system monitoring devices which rely on optics and thus require clean windows to function effectively [5–8]. Sub-aquatic systems include vehicles, tidal power turbines, communication links, pipelines and oil well heads. For example, deep sea oil well trees (assemblies of fittings to regulate oil flow) are designed to have a lifetime of 30+ years without servicing, so the windows on the monitoring equipment need to be remotely or autonomously cleaned. Anyone who has ever cleaned a greasy frying pan in the kitchen sink will be aware of the difficulty in trying to wipe oil clear of a surface: oil is immiscible in water, making mechanical wiping and ultrasonic agitation not particularly effective for subaquatic application [9]. Photo-active, catalytic, self-cleaning coatings, however, may be effective under water, as they do not displace the contaminant, but degrade it [10,11].

Sol-gel technology is here demonstrated as the fabrication method for a number of reasons. Primarily it may be easily and affordably produced on three-dimensional surfaces and crystallized into metallic ceramic such as $TiO_2$ [12]. The chemical synthesis facilitates many alternative compositions or dopants [12]. Additionally, micro and nanopatterning is readily achievable via imprinting to alter the surface topography [13].

The literature suggests a number of mechanisms which may enhance the self-cleaning effect of anatase [14–16]. Here we explore a few of these options to evaluate, on a qualitative level, whether any cleansing enhancement may be obtained under water. Increasing the surface area is a trivial means of increasing the rate of any physiochemical reaction, but it is not trivial to assume that micro-patterning will increase the area enough to provide a visible enhancement. Therefore, increased surface roughness is here evaluated against a planar control. Another means of potentially enhancing the self-cleaning effect is to incorporate elemental dopants. Dopants can effect hydrophobic behavior, bio-compatibility and the rate of the photocatalytic effect (photon induced electron release). This entirely depends on the dopant and type of contaminant. Metal ions may be characterized by oxidation state. A positive oxidation state dopant, such as Manganese (+7), is designed to shift the electron/hole ratio in a dielectric, such as $TiO_2$ (+4), in favor of electrons, and, thus, theoretically increase the photocatalytic effect [17,18]. However, electron/hole recombination will be rapid so the relative dirt attack time will be reduced [19]. On the contrary, metal ions such as gadolinium can only be found with oxidation states lower than $TiO_2$ (+4), so such a dopant is expected to reduce the photocatalytic effect [20,21]. But it does have space in its electron shell to accommodate extra electrons. By harboring an electron, positive charges are left to attack organic surface matter. In addition, with reduced photo catalysis comes greater hydrophobicity, which may be beneficial for inhibiting contamination and easing removal of dirt in the subaquatic conditions [22]. $SiO_2$ has the same oxidation state as $TiO_2$ (+4) but $SiO_2$ is a known adsorbent of hydrocarbons [23]. Previously, sol-gel synthesized compounds of $SiO_2$ and $TiO_2$ have proven more effective in aqueous mixtures than $TiO_2$ alone, because the $SiO_2$ has increased the concentration of organic species around the $TiO_2$ molecules [23]. Hence, $SiO_2$ sol-gels shall also be synthesized in the same manner as the $TiO_2$ version, and subsequently mixed to determine whether the inclusion of a known adsorbent will enhance the self-cleaning effect in this study. The trade-off of doping, regardless of positive or negative oxidation state ions, is application specific as, ultimately, this performance tuning comes down to the specific experimental conditions: primarily the type of dirt, method of fouling and wavelength of the light source concerned. Therefore, several variants are tested on a qualitative level for a general enhancement to the cleansing process.

Due to the inherent challenge the removal of greasy contamination poses for subaquatic surfaces, our evaluation is predominantly focused on this form of contaminant. In order to quantify how effective the self-cleaning windows are at removing greasy contamination, water contact angle (WCA) analysis is performed. WCA provides a relative metric for the self-cleaning capacity of each window, as it describes the wettability of surfaces. Particles of contamination on a flat surface will induce air pockets at the water droplet interface, and the Cassie-Baxter theory explains that this will induce a higher WCA. Grease and oil are known to be immiscible with water, so such contaminants additionally induce a high WCA. Thus, clean surfaces have a low WCA compared to those fouled with greasy contamination, and a self-cleaning surface will restore a low WCA. Large changes in WCA are indicative of high-performance self-cleaning windows. The coating most effective at degrading greasy contamination is further evaluated for compatibility as an anti-biofouling surface. For biological and crude oil contaminants the self-cleaning effect is evaluated by analysis of optical transmission strength.

## 2. Methods

Some 3 mm thick, 25 mm diameter sapphire windows were sourced from UQG Optics to use as substrates. Substrates were spin coated with sol-gels at 9 krpm for 10 s. All chemicals were sourced from Sigma-Aldrich (Irvine, Scotland, UK). The titania sol-gel base solution (molar ratio: $TiNC_{46}H_7O_{10}$) is prepared by mixing 0.96 mL of diethanolamine with 5.54 mL of 1-hexanol and 0.10 mL of deionized

water. Diethanolamine is a solid at room temperature, so the source bottle requires heating above the melting point of 28 °C, before a decantation may be made. The mixture should be vigorously stirred for 10 min, before slowly adding 3.40 g of titanium butoxide while stirring. To synthesize the silicon counterpart, the 3.40 g of titanium butoxide was replaced by 0.34 g of tetraethyl orthosilicate. The two sols were then mixed together in a ratio of 1:1 to establish a Ti:Si ratio of 10:1. 10% Si inclusion has previously been shown to be optimum for enhancing the photocatalytic reaction [22]. Some powder chemicals were sourced and dissolved into the $TiO_2$ sol:

- Anatase crystals ≤ 3 μm Ø (to examine increased particle size);
- Gadolinium acetate (to examine a lower oxidation state dopant);
- Manganese acetate (to examine a higher oxidation state dopant).

All of the above dopants were dissolved into the sol-gel via stirring until saturation at room temperature. With the exception of the anatase crystal dopant these were then passed through a 0.2 μm filter, to ensure particle size was not a contributing factor in the self-cleaning analysis. In addition to chemical variations, a micropatterned PDMS stamp was imprinted onto a $TiO_2$ sol coating to produce 100 μm diameter, 80 nm deep circular protrusions. After spin-coating (and imprinting), samples were annealed in a furnace at 500 °C for 4.5 h to realize their corresponding (circa 80 nm thick) ceramics.

Each sample was mounted as the window of a watertight lens tube featuring a 365 nm, 2.9 W LED from LED-Engin, as shown below in Figure 1. All the window samples were contaminated using a thumbprint as contamination. A natural thumbprint was chosen as the source of greasy contamination due to its intrinsic visual pattern and heterogeneous composition (a complex emulsion of organic and inorganic residue predominantly proteins and sodium chloride salt respectively) [24]. Using an Attension Theta optical tensiometer, three sessile drop WCA measurements were taken from each contaminated sample before and after one day of UV exposure in a tank of flowing tap water.

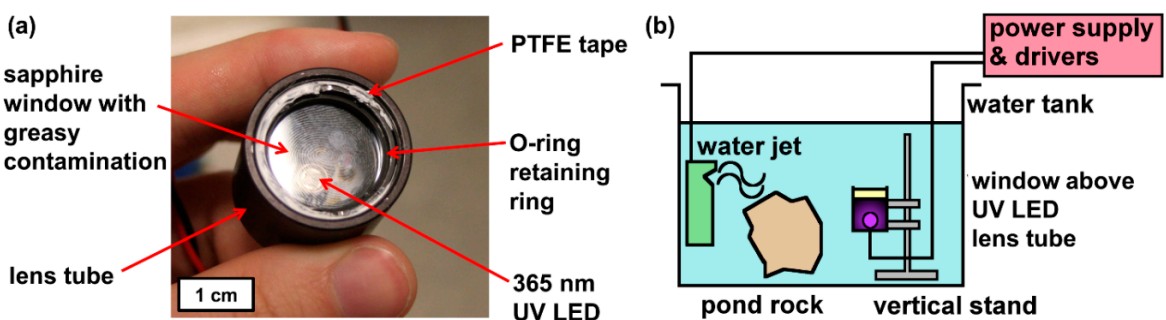

**Figure 1.** (**a**) Annotated photograph of submergence lens tube set-up. (**b**) Schematic of the tank set-up (note: pond rock only present for biofouling tests to ensure rich bioactivity in pond water).

After determining the most effective of the tested coatings with a thin greasy fingerprint, a more appropriate contaminant is applied: thick, crude oil from North Sea Troll B. The crude oil was spin coated onto the windows at 3k rpm to ensure intra-batch coating consistency. Optical transmission power of 365 nm UV light was measured with a Thorlabs PM100 USB power meter, equipped with a S302C thermal power sensor capable of measuring 190–2940 nm radiation to 5% certainty. Measurements were taken before and after contamination, and then again after 21 h of UV exposure. The lens tubes were mounted vertically inside the tank, as shown in schematic Figure 1b, to avoid the oil slumping to the edge of the window.

Additionally, the optimum coating was assessed for compatibility as an anti-biofouling interface. The windows were mounted on the same lens tubes as before and submerged in a tank containing pond water. Bacteria grew in the tank and on the tested windows. Spectral transmission measurements were made, using a Jenway 6700 Spectrophotometer, on samples which had been under constant UV exposure in the tank for 45 h, and compared to controls that had not been exposed. The samples which

were not exposed for the first 45 h were subsequently exposed for the same duration, to confirm that self-cleaning was occurring, and not just inhibition of microbe formation.

## 3. Results and Discussion

Initially, the $TiO_2$ anatase base material was tested in a lens tube in an air atmosphere to confirm that the material was functional prior to submergence in water. The $TiO_2$ base material proved highly effective in removing thumbprint contamination, as shown in Figure 2, and it almost completely eradicated a thumbprint within 24 h of UV exposure in air, whereas the same treatment on the non-coated sapphire left the thumbprint visible to the naked eye.

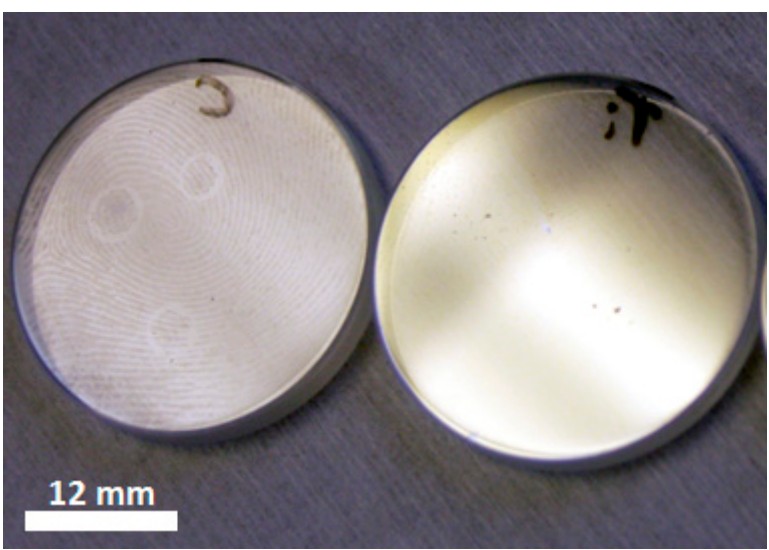

**Figure 2.** Photographic comparison of the fingerprint removal study on a sapphire window (**left**) and an anatase coated sapphire window (**right**), following treatment with 365 nm UV exposure for 24 h in air.

Table 1 documents the WCA of the various tested coatings after fouling, and after 24 h of submergence in a tank of flowing tap water with constant 365 nm UV exposure. For comparison the WCA measurements of the $TiO_2$ coating from the air experiment are also documented in Table 1. Boxes are colored with a traffic light system, where red displays a poor result, orange an indifferent result, yellow a good result, and dark green the best result. The difference (delta) between the contaminated and exposed samples is also tabulated. It can be seen from an examination of Table 1 that the standard $TiO_2$ coating performed the best. The contact angle was reduced to 22.66 degrees (on average) under flowing water. This was even better than it had performed in air, indicating that the flowing water has aided removal of debris from the sample surface.

**Table 1.** Average water contact angle (WCA) of the tested coatings before and after 365 nm UV exposure for 24 h submerged in flowing tap water.

| WCA Table (Degree) | Sapphire (Control) | TiO$_2$ (in Air Control) | TiO$_2$ | Micro Patterned | Anatase Doped | Gd Doped | Mn Doped | Si Doped |
|---|---|---|---|---|---|---|---|---|
| contaminated | 80.82 | 70.39 | 68.79 | 40.22 | 67.01 | 65.03 | 64.01 | 66.73 |
| 24 h UV exposure | 77.34 | 31.14 | 22.66 | 55.38 | 55.74 | 34.52 | 77.77 | 63.49 |
| delta | 3.48 | 39.24 | 46.13 | −15.2 | 11.27 | 30.51 | −13.8 | 3.24 |

The reason that the anatase doped and micro patterned surfaces were determined to be ineffective was because the increased roughness of the surfaces physically traps debris between topographical features. The manganese doped sol-gel became more contaminated after submergence, suggesting that,

under these conditions, the electron/hole recombination rate is so rapid the electron holes do not have enough time to effectively attack the surface contamination, rendering the coating ineffective and free to be fouled. Post treatment, the WCA of the manganese doped coating is seen to be almost identical to the non-active sapphire control sample. These two samples were the worst performing surfaces in the test. Gadolinium on the other hand has lower oxidation states than $TiO_2$, and, in contrast to the Mn doped sol-gel, the Gd doped version performed well. It achieved second best in the submergence study but, curiously, was not as effective as intrinsic $TiO_2$. This result suggests that the electron vacancies incorporated with the Gd dopant inhibit the photocatalytic effect of the generic titania coating, as more energy is required to excite the electrons over intrinsic $TiO_2$. The combination of $SiO_2$ and $TiO_2$ behaved similarly to the non-active sapphire control (delta = ~3.3 degree). This suggests that the surface of the hybrid coating may have been dominated by the non-active $SiO_2$ ceramic. To test whether a thin non-active surface film would inhibit the $TiO_2$ photocatalytic behavior, 5 nm of Au was evaporated onto the generic sol-gel coating. It too failed to elicit the self-cleaning effect (results included in the Supplementary Materials Pages S4), confirming that the $TiO_2$ coating requires to be in direct contact with the organic contaminant in order to function.

After establishing that the generic $TiO_2$ anatase coating (as characterized by X-ray photoelectron spectroscopy, Raman analysis and atomic force microscopy, included in the Supplementary Pages) was the most effective of the fabricated self-cleaning coatings, a more profound test was performed to cleanse crude oil from the surface, so as to restore the transmission of light. After 21 h of 365 nm UV exposure, the $TiO_2$ coating had induced degradation in the crude oil, to the extent that power transmission was restored to within 10% of its clean surface value. Figure 3 displays photographs of the $TiO_2$ sol-gel coated window before and after UV treatment. The brown colored dead crude oil film appears to have undergone cold combustion following the treatment, as dry, condensed, white residue remains on only a partial area of the window. Figure 4 plots the transmission power of 365 nm UV light through the window before contamination, after contamination and after 21 h of exposure in flowing tap water. This is a marked enhancement over not using the anatase coating. The control for Figures 3 and 4 are non-coated sapphire windows illuminated with UV exposure for the same duration.

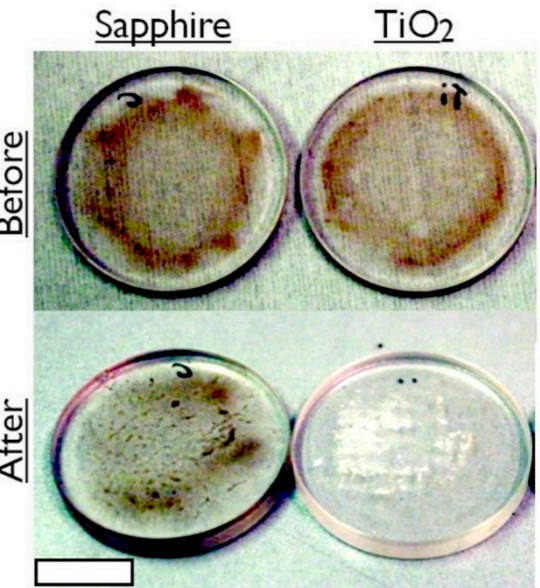

**Figure 3.** Photographs of a standard sapphire window and a UV-exposed $TiO_2$ coated window spin coated at 3k rpm with dead North Sea crude oil (API 35), before and after 21 h of submergence in flowing water. Scale bar = 10 mm.

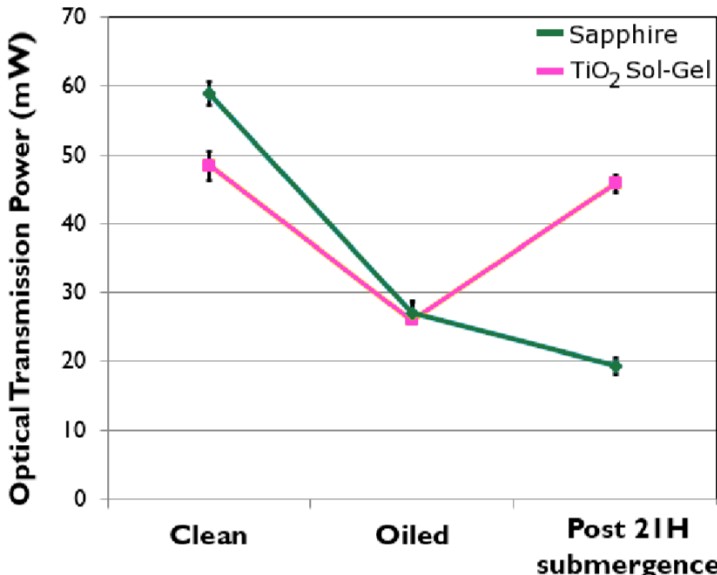

**Figure 4.** Optical 365 nm transmission power through samples before and after contamination and again after 21 h of submergence. Note: the sapphire control sample was not exposed during the submergence, as it represents the current non-photoactive window system.

It may be observed from Figure 4 that prior to contamination the $TiO_2$ sol-gel induces a reduction of around a sixth of the transmitted power; however, the oil is so opaque that both the sapphire control and sol-gel coated sample deliver near identical power levels once contaminated. Imperatively, the photo-active sol-gel coating restored 90% of the power after 21 h, whereas the control sapphire, which was not illuminated, lost a total of 68% by the end of 21 h submerged in water.

The photo-active titania coating was also evaluated for effectiveness in retaining spectral transmission in a bio-active environment. It was discovered that, although primarily designed to degrade greasy contamination, the coating also functions effectively for cleansing microbe fouling. Following 45-h submergence in flowing bio-active pond water, Figure 5 compares the optical transmission strength of the titania-coated windows with and without UV illumination. Under constant illumination (turquoise trace) the window transmits 10% more than the dark control (orange trace), indicating biofouling is inhibited for illuminated samples. To demonstrate that the illumination process does not simply inhibit bio-film formation, and to further demonstrate that the window actively self-cleans such contamination underwater, the contaminated control window was subsequently illuminated (blue trace). The window did indeed clean the biofouling from the surface as, Figure 5 shows, transmission strength was restored by up to 8%.

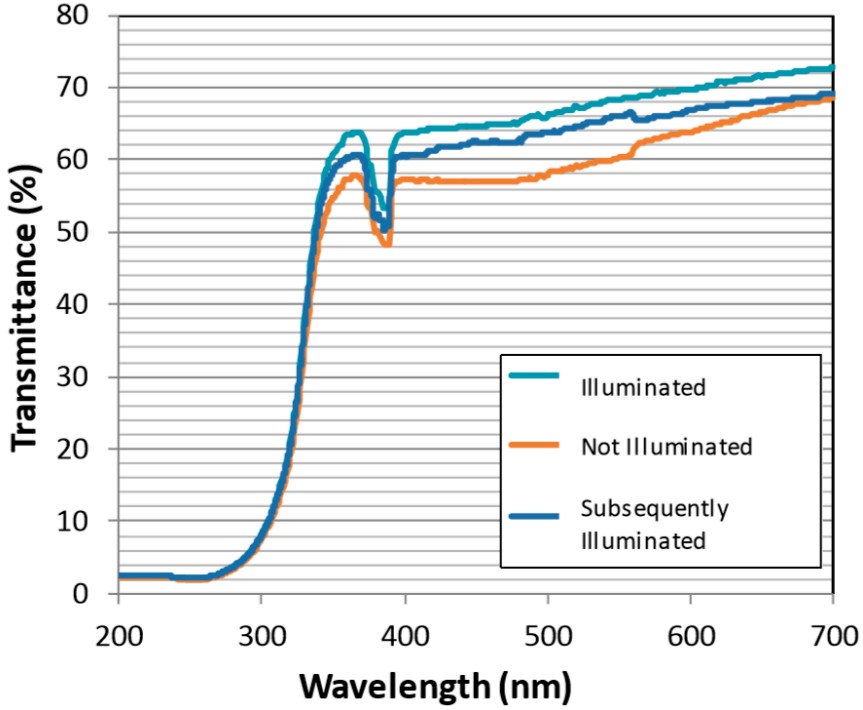

**Figure 5.** Spectral transmission of the photo-active coating after submergence in a bio-active environment. The 'Illuminated' sample was exposed constantly for 45 h (turquoise). The 'Not Illuminated' sample was in submergence for 45 h (orange), before being 'Subsequently Illuminated' for 45 h (blue).

## 4. Conclusions

Sol-gel derived $TiO_2$ in the anatase polymorph has been shown to be an effective self-cleaning coating under subaquatic conditions. Dopants of higher, lower and equal oxidation states were all tested. With the inclusion of dopants, there is a tradeoff between the energy required to activate the process and the reaction duration. In this instance attempts at doping only compromised the cleansing effect. Both increasing the anatase particle size and surface patterning failed to increase the cleansing effect. The reason these two samples were both determined to be ineffective is because the increased surface roughness hinders the release of debris. Despite attempts to enhance the self-cleaning performance of sol-gel derived titania proving ineffective, the intrinsic coating worked highly effectively. Following worst case scenario contamination of the titania coated window, with dead North Sea Troll B crude oil, optical power transmission was restored to 90% after 21 h. In the absence of the photo-active sol-gel derived titania coating, sapphire windows suffered a 68% loss in optical power under the same test conditions. In addition, biological contamination may also be cleansed from the surface using the titania coating making it suitable for a range of subaquatic applications, where oil or biofilms are known contaminants.

**Supplementary Materials:** The following are available online at http://www.mdpi.com/2073-4352/10/5/375/s1, Figure S1: XPS spectrum for titania sol-gel annealed at 500 °C; Figure S2: Recorded Raman spectrum for titania sol-gel annealed at 500 °C, conforming to the known anatase polymorph spectrum. Figure S3: Atomic force microscopy (AFM) surface profile scan of a $3.0 \times 3.0~\mu m^2$ area of a synthesized anatase $TiO_2$ coating; Figure S4: Visual reference between the tested windows following sub-aquatic cleansing tests; Table S1: Water contact angle measurements before and after UV exposure for a sapphire control, $TiO_2$ sol-gel coating and a $TiO_2$ sol-gel coating with 5 nm of Au evaporated on to the surface.

**Author Contributions:** Conceptualization, N.G., D.M. and G.K.; methodology, A.I.M.G. and N.G.; validation, A.I.M.G. and N.G.; formal analysis, A.I.M.G.; investigation, A.I.M.G.; resources, D.M., G.K. and N.G.; data curation, A.I.M.G.; writing—original draft preparation, A.I.M.G.; writing—review and editing, A.I.M.G., D.M. and N.G.; visualization, A.I.M.G.; supervision, N.G.; project administration, N.G. and G.K.; funding acquisition, G.K. All authors have read and agreed to the published version of the manuscript.

**Funding:** The authors would like to acknowledge the funding provided by FMC Technologies Inc to carry out this research. AIMG was funded by EPSRC (EP/P505534/1) 63677.

**Acknowledgments:** We acknowledge the James Watt Nanofabrication Center for fabrication work.

**Conflicts of Interest:** The authors declare no conflict of interest. The funders had no role in the design of the study; in the collection, analyses, or interpretation of data; in the writing of the manuscript, or in the decision to publish the results.

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
