# Peer review of "Sol-Gel Coatings for Subaquatic Self-Cleaning Windows"

_crystals, doi:10.3390/cryst10050375_

Round 1

Reviewer 1 Report

The manuscript describes a comparison between several sol-gel coatings, doped and undoped, for photocatalysis of surface contaminants on sub-aquatic windows. The paper is interesting and a useful contribution to the field - particularly for water-based optical instrumentation. I have some comments and questions I feel may help clarify some points:

1) It would be useful to mention the molar ratio of the sol-gel mixture;

2) The PDMS stamp gives an 80nm deep well - what is the overall film thickness?

3) For depositing the greasy fingerprint - what was the substance, and was it an index finger pressed into the substance then immediately pressed onto the window? Forensics papers have shown there is significant different between fingers, number of pressings etc for finger mark impressions;

4) How long was the tube submersed in the pond water? Was there anything else in the tank?

5) Figure 2 - it would be useful to have a false-colour version of the photo or something next to the original to better distinguish the differences;

6) If the oil is spin-coated on the horizontal plane, does the oil slump to to the bottom edge of the substrate when placed in a presumably vertical plane for measurement?

7) The section regarding the experimental procedure leading to Figure 5 was not clear to me, particularly with respect to the non-illuminated control. This section should be expanded.

Author Response

1) It would be useful to mention the molar ratio of the sol-gel mixture;

The molar ratio of the titania sol-gel mixture is now referenced in the methods section.

2) The PDMS stamp gives an 80nm deep well - what is the overall film thickness?

We would ideally like to answer this question via ellipsometry analysis but current government advice in the UK has prevented lab access at this time due to COVID-19. We do know from analysis of spin coating the same sol-gel with the same spin settings on Si and metal substrates that the post-annealed depth is typically 80 nm prior to imprinting, and is now referenced as such in the manuscript.

3) For depositing the greasy fingerprint - what was the substance, and was it an index finger pressed into the substance then immediately pressed onto the window? Forensics papers have shown there is significant different between fingers, number of pressings etc for finger mark impressions;

The method section now updated: A natural thumbprint was chosen as the source of greasy contamination in Figure 2 due to its intrinsic visual pattern and heterogeneous composition (a complex emulsion of organic and inorganic residue predominantly proteins and sodium chloride salt respectively)26.

4) How long was the tube submersed in the pond water? Was there anything else in the tank?

The lens tube was submerged for 45 hour in flowing bio-active pond water, there was also a porous rock from the pond present to ensure bio-activity. The text has been updated to clarify this.

5) Figure 2 - it would be useful to have a false-colour version of the photo or something next to the original to better distinguish the differences;

The image has been replaced with a higher contrast image.

6) If the oil is spin-coated on the horizontal plane, does the oil slump to to the bottom edge of the substrate when placed in a presumably vertical plane for measurement?

Oil was kept in the horizontal plane during static (not flowing) submergence and this is now clarified in the text.

7) The section regarding the experimental procedure leading to Figure 5 was not clear to me, particularly with respect to the non-illuminated control. This section should be expanded.

Section now re-written to clarify: Following 45 hour submergence in flowing bio-active pond water, Figure 5 compares the optical transmission strength of the titania-coated windows with and without UV illumination.  Under constant illumination (turquoise trace) the window transmits 10% more than the dark control (orange trace), indicating biofouling is inhibited for illuminated samples. To demonstrate that the illumination process does not simply inhibit bio-film formation and to further demonstrate that the window actively self-cleans such contamination underwater, the contaminated control window was subsequently illuminated (blue trace). The window did indeed clean the biofouling from the surface as, Figure 5 shows, transmission strength was restored by up to 8%.

Reviewer 2 Report

This paper reports a sol-gel coating for a subaquatic self-cleaning sapphire windows application. The topic is interesting and useful, and the procedure was properly described but the materials synthesized were not properly characterized, if the story that is more complete can be presented, the paper could be considered for publication.

In general, I think the research presented here is of interesting, and the parts of application of TiO2 coating on the sapphire windows and the performance are quite good written, however, the story is not complete, we don't have a picture of the material itself, how is the coating, the characteristics of crystal structure, micro morphology, and coating quality etc.

Author Response

we don't have a picture of the material itself, how is the coating, the characteristics of crystal structure, micro morphology, and coating quality etc.

The synthesized coating is now evaluated by XPS, Raman and AFM in the supplementary pages. Visually there are few features to observe on the surface, as may be deduced from the AFM profile. COVID-19 is presently inhibiting lab access so additional analysis is not possible at this time.

Reviewer 3 Report

The article deals with a very well known and studied subject. Unfortunately, in its present form it does not bring any news compared to literature.
Moreover, the form in which it is written is approximate and should be reviewed in an important way. There are also important shortcomings in the introduction especially on the self-cleaning part (Cassie Baxter theory, contact angle, etc.).

If the experimental and characterization setup was improved, more quantitative and not only qualitative data would certainly be of interest.

Regards

Author Response

The article deals with a very well known and studied subject. Unfortunately, in its present form it does not bring any news compared to literature. 

To the best of our knowledge, there are no previous reports in the literature demonstrating the reported self-cleaning properties under water.

There are also important shortcomings in the introduction especially on the self-cleaning part (Cassie Baxter theory, contact angle, etc.).

Methods now updated: Water contact angle (WCA) provides a relative metric for the self-cleaning capacity of each window as it describes the wettability of surfaces. Particles of contamination on a flat surface will induce air pockets at the water droplet interface; Cassie-Baxter theory explains that this will induce a higher WCA. Grease and oil are known to be immiscible with water, so such contaminants additionally induce a high WCA. Thus, clean surfaces have a low WCA compared to those fouled with greasy contamination and a self-cleaning surface will restore a low WCA. Large changes in WCA are indicative of high-performance self-cleaning windows. Using an Attension Theta optical tensiometer, three sessile drop WCA measurements were taken from each contaminated sample before and after one day of UV exposure in a tank of flowing tap water.  

Moreover, the form in which it is written is approximate and should be reviewed in an important way. If the experimental and characterization setup was improved, more quantitative and not only qualitative data would certainly be of interest.

The nature of this study is entirely fundamental in identifying whether (a) self-cleaning window technology functions under water, (b) relevant sub-aquatic contamination may be actively degraded and (c) if the self-cleaning effect may be enhanced via chemical or physical modification. Despite the qualitative nature of this study, all of the WCA analysis was performed in triplicate for statistical significance to determine the optimum coating from the presented options. This optimum coating was subsequently reproduced across multiple windows for scientific rigor. We believe the testing is suitably robust for publication.

Round 2

Reviewer 2 Report

The topic is still interesting to me. And the author did some characterizations for the film, it still would be good if the properties of the materials itself could be provided, I understand during the COVID 19 lockdown, it is challenging for providing new experiments, therefore, I leave this to the editor to decide. For me, the paper could be accepted better with more characterizations.

Author Response

Following the last set of revisions we have characterised the effective self-cleaning window material by providing proof of the TiO2 composition via XPS analysis, the anatase crystal phase via Raman spectroscopy and the surface structure via AFM. All of the aforementioned analysis is included in the supplementary pages.

We do not currently have access to laboratories due to COVID-19 UK lock-down, so further analysis is not possible at present but we believe we have provided sufficient material characterisation for publication.

Reviewer 3 Report

Dear Authors,

I appreciate the work on paper because in this version is more complete and, for me, clear. I have only some suggestion to improve the form.

I appraciate the improvment and the detail about WCA but i think that could be oppurtune to put some reference and explanation about the self cleaning in the introduction. Is more appropiate to read this information in intruduction section with respect to in methods section, here you can put some notes only.

For me it's not very clear the set up of the submerged experiments: sample position, illumination and so on. I suggest to draw a sketch of it.

At the end of line 103 there is a "26". It is a reference or an error?

Author Response

The reviewer is quiet correct in that the background literature regarding WCA should appear in the introduction, so is now located there.

Figure 1 now includes a schematic to illustrate the submergence test set-up as suggested.

Reference 26 is now formatted appropriately.